# The Resilience of the Renewable Energy Electromobility Supply Chain: Review and Trends

**Alma Delia Torres-Rivera** [1,*] , **Angel de Jesus Mc Namara Valdes** [2] **and Rodrigo Florencio Da Silva** [3,*]

1   UPIEM del Instituto Politécnico Nacional, Gustavo A. Madero 07738, Mexico
2   Urban Mobility Engineering of UPIEM del Instituto Politécnico Nacional, Gustavo A. Madero 07738, Mexico; amcnamarav@ipn.mx
3   ESIME Ticomán del Instituto Politécnico Nacional, Gustavo A. Madero 07738, Mexico
*   Correspondence: atorresri@ipn.mx (A.D.T.-R.); rflorencio@ipn.mx (R.F.D.S.)

**Abstract:** Electromobility has been crucial in mitigating transport emissions and meeting reduction targets. From this context, this literature review's main objective is to analyze the resilience of the electromobility supply chain that integrates renewable energy sources. This literature review focuses on the resilience of the electromobility supply chain and how it can incorporate renewable energy sources. The central argument is that the success of the supply chain depends on its ability to resist, adapt, and recover from disruptions that affect operations. We comprehensively review current knowledge in three stages: identifying critical components of resilience, highlighting challenges and opportunities for risk mitigation, establishing strategic alliances, and synthesizing vital issues, trends, and emerging areas that require further research. The findings emphasize the importance of improving supply chain resilience for sustainable transportation and environmental preservation under five guidelines: emergency preparedness, monitoring and evaluation, sustainable practices, maintenance of essential services, and prevention of operational disruptions.

**Keywords:** resilience; supply chain; sustainability; electromobility; renewable energies; review





## 1. Introduction

The demand for environmentally sustainable transportation has increased due to rising energy consumption and greenhouse gas emissions. Electromobility is emerging as a viable solution to tackle this challenge by integrating renewable energies into the supply chain. However, the transition towards sustainability poses several challenges for the actors involved, especially during the pandemic, highlighting the need for resilience.

The COVID-19 pandemic has caused significant changes in the global supply chain, leading companies to diversify their suppliers and adopt advanced technologies such as artificial intelligence and the Internet of Things [1]. Supply chain management aims to ensure the ability to withstand and quickly recover from unexpected disruptions. Stakeholders work collaboratively to anticipate risks and implement measures that enable a stronger, more flexible, and more agile sustainable supply chain.

Integrating renewable energy within the supply chain can contribute to sustainability within each link by reducing greenhouse gas emissions and minimizing the industry's environmental impact. In the case of producing electric vehicles and charging infrastructure, it implies the acquisition of raw materials, logistics, and waste management. Therefore, the sustainable management of the electromobility supply chain through integrating renewable energies is crucial for zero-emission transport. Therefore, sustainable managing the electromobility supply chain by integrating renewable energy is essential to ensure sustainability and efficiency.

The electromobility supply chain is one option to reduce greenhouse gas emissions. The adoption of wind, solar, hydroelectric, and green hydrogen energy in electric vehicles requires joint action between industries, government, and consumers in the development

of infrastructure, policies, and laws for change, and the benefit of electromobility is to contribute to the reduction of dependence on fossil fuels [2].

Electromobility is the product of the convergence of vehicles that run entirely or partially on electricity, either from fossil sources or renewable energy sources. The electricity used to charge the batteries from non-renewable sources or fossil fuels is not emission-free. These electric vehicles can be cars, buses, motorcycles, bicycles, and other means of transportation. Ground transportation consumes about 81% of the fossil fuel demand of the entire sector worldwide, making the path to decarbonization a priority in North America [3]. The goal for 2050 is that the whole vehicle fleet will comprise zero-emission electric vehicles.

Electric vehicles can be powered by electricity generated from renewable sources, such as solar panels or wind turbines, allowing for cleaner and more sustainable transportation. In addition, using renewable energy in the transport sector can also help diversify and decentralize energy production, fostering local job creation and energy autonomy. Hence, the generation and utilization of renewable energy, such as solar and wind power, necessitate implementing energy storage systems and intelligent management of lithium battery charging to optimize the utilization of renewable energy. Lithium batteries are characterized by their high energy density and ability to provide sufficient autonomy. Moreover, these batteries can be recharged, enabling the reuse of stored energy.

The expansion of electromobility and the growing demand for lithium batteries pose several supply chain challenges. For example, ensuring a stable and long-lasting supply of lithium batteries requires significant raw materials, such as lithium, cobalt, nickel, and manganese. This, in turn, enhances the overall efficiency and performance of batteries. It is worth noting that lithium is an abundant natural element that can be sustainably extracted for battery manufacturing. Still, the stakeholders are working on adopting sustainable practices into their operations and processes under organizations to minimize potential adverse environmental impacts and generate social value [4–7]. The push for electromobility represents a significant opportunity but also demands careful planning and management to overcome the challenges associated with the supply chain.

Other aspects of the electromobility supply chain are linked to the technical diversity available for motor design and power electronics with different voltage levels in parallel and coaxial shaft systems using silicon carbide as the semiconductor material. For manufacturers and suppliers alike, the question arises as to whether silicon carbide improves efficiency, increases power density, and provides superior performance in high-voltage systems, thereby contributing to improved performance and reliability autonomy of electric vehicles.

Sustainability as a condition [8–10] is the ability to maintain a functional system through complete tasks to solve environmental problems and drive economic development and social progress [11]. Based on this reasoning, sustainable management has four fundamental aspects: (a) the generation of profitability; (b) the promotion of respect for the environment as an organizational value; (c) social responsibility; and (d) technological innovation [12]. Therefore, it is influential in trading strategies. The contribution of Hart in the year 1995, stands out, which from an organizational perspective, focuses on natural resources, while Freeman, 1984, uses stakeholder analysis to incorporate sustainability in supply chains [13].

Thus, sustainability promotes the minimization of pollution and the overexploitation of non-renewable resources and favors social responsibility actions to assume the negative externalities it generates for the community. Sustainability is a philosophy that aims at the quality of life of people and the planet, for which it must ensure the economic retribution of the factors of production. Otherwise, the quality of life would not be sustainable over time.

In most parts of the world, the automotive industry is developing the last generation of drives with an internal combustion engine. This will give a solid impetus to electromobility. The COVID-19 pandemic was one of the main reasons for the further acceleration of electromobility, which is entering the decisive phase in which it will replace combustion

engines for electric mobility. The key to success will be the costs and the attainable range for a given battery size.

The electromobility supply chain with renewable energies involves links and actors working together to produce and distribute electric vehicles and their components. The main links and actors in the supply chain:

- Raw materials: Electric vehicle manufacturers need various materials to produce batteries, motors, and other components. Some critical materials include lithium, cobalt, nickel, graphite, aluminum, and copper;
- Component suppliers: Electric vehicle manufacturers rely on a network of suppliers that provide key components such as batteries, motors, and electronic controllers;
- Electric vehicle production and distribution through a network of players are responsible for the sale and after-sales service;
- Adopting electric vehicles requires adequate charging infrastructure to enable drivers to charge them conveniently and quickly. In other words, it involves installing charging stations in homes, parking lots, highways, and other vital locations;
- Renewable energy: Renewable energy, such as solar and wind power, is essential for transitioning to sustainable electromobility. Using renewable energy sources to charge electric vehicles reduces the carbon footprint of electric mobility.

The renewable energy electromobility supply chain involves a complex network of actors and links working together to produce and distribute sustainable electric vehicles and reduce dependence on fossil fuels [14]. In this way, the management of the electromobility supply chain poses two problems that significantly affect its operation. The first is adopting sustainable practices in electromobility, an issue that can substantially affect the continuity of companies. The second problem is associated with the capacity of the supply chain to resist, adapt, and recover from disturbances or unexpected events. Numerous authors and researchers in the field have dealt with this issue, and they all reached a consensus that resilience is determined by the agility, flexibility, and resistance of processes and the continuity of operations in the face of changes in the context of globalization.

Figure 1 illustrates the main stages of the sustainability supply chain for producing electric vehicles. The first stage concerns procuring raw materials to manufacture components: lithium, cobalt, graphite, steel, aluminum, and copper. For the manufacturing stage, there are essential components that must be considered for the manufacture of electric cars, including batteries, electric motors, battery charging, management systems, and the chassis. The next stage is distribution, where technology and communication play an essential role in ensuring that the components and the finished vehicles are on time and ready to be used or sold. After-sales service has been added because electric cars require infrastructure for battery charging, maintenance, and specialized spare parts for the vehicle to be used by the end customer.

This study aims to analyze the resilience of the electromobility supply chain and the integration of renewable energies to provide a solid knowledge base to develop effective strategies and make informed decisions in the design, management, and optimization of supply chains in the electromobility sector. From the analysis of the results, four fundamental principles stand out to guide the direction of electromobility supply chains with the integration of renewable energies: (i) develop emergency preparedness; (ii) operational continuity plans; (iii) carry out monitoring, surveillance, and risk assessment; (iv) implement sustainable practices for inventory management and prevention of shortages; and (v) a strategic action plan before operating interruptions.

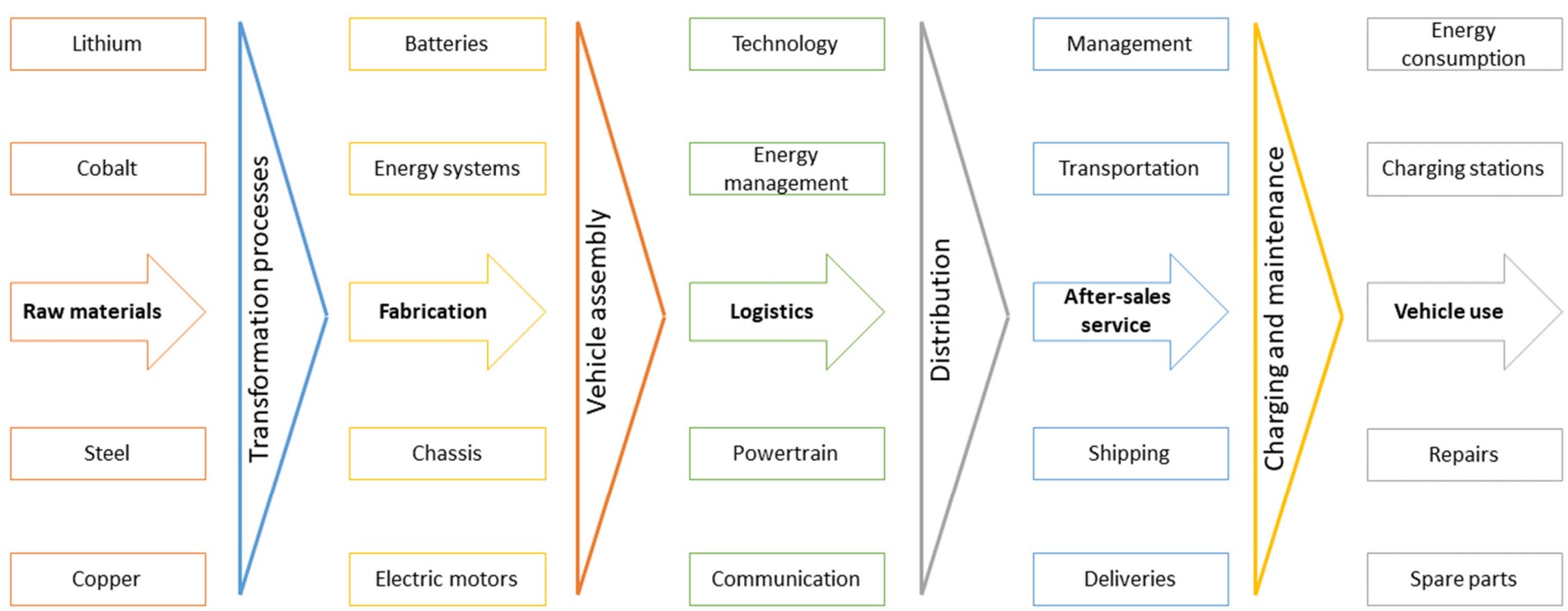

**Figure 1.** Electromobility supply chain management.

Several reasons have led to this objective. The first stems from the fact that there are various investigations on the electromobility supply chain to facilitate management. Some studies conclude that sustainability is integrated into strategic planning with an economic focus and in the forward flow. At the same time, the resilience capacity is evaluated by constructing scenarios from the risk assessment perspective [15]. However, only some have analyzed the relationship between sustainability, resilience, and renewable energy.

Secondly, recent studies have focused attention on the micro-organizational properties that can favor supply chain management (interaction between actors, collaboration, and integration of the tasks they develop), or in intra-organizational relations, mainly from the theoretical point of view that the resilience of organizations could be developed through the planning and implementation of lean production, continuous improvement practices, the flexibility of their processes, and the agility to respond to interruptions or changes in the environment [16]. As a result, organizations could speed up the recovery process after a disruption anywhere in the supply chain.

The results show that the resilience of the electromobility supply chain globally is associated with implementing strategies and practices that allow an organization to adapt and recover quickly from interruptions and unexpected events. Supplier diversification, redundancy, collaboration, risk planning, and the use of technology are critical elements in strengthening supply chain resilience.

This paper provides a novel conceptual contribution that attempts to answer these questions by describing some critical components of electromobility supply chain resilience. First, the document defines the dimensions of resilience that affect electromobility that transitions to renewable energy. More specifically, the paper looks at the nexus of electromobility with renewable energy integration and supply chain management.

To do this, it is essential first to understand the electromobility supply chain that uses renewable energy from the characteristics that revolve around resilience. Secondly, the methodology used is exposed. Third, the principal results and their discussion are shown. Finally, the conclusions, implications, and future lines of research derived from the study are offered.

## 2. Methodology

The literature review methodology to analyze the resilience of the electromobility supply chain and the integration of renewable energy involved the following steps:

To identify the articles, a search was performed in several databases (Web of Science, EBSCO-host, ProQuest, Emerald, and Science Direct). The investigation was performed using the following combinations of terms: ("Supply Chain" OR "Value Chain" OR "Logistics" OR "Logistics Network") AND ("Resilience" OR "Resilience" OR "Adaptive Capacity" OR "Disruption") AND ("Electric Vehicle" OR " EV" OR "electric car" OR "electric transportation") AND ("Renewable energy" OR "clean energy" OR "sustainable energy" OR "renewable energy") in the title, abstract and/or keywords. The search was performed using Harzings' Publish or Perish software (2016 version) for Windows to retrieve the number of citation metrics, including the number of articles, total citations, and h-index. Reference lists in retrieved studies were also checked.

The inclusion criteria that were used to retrieve the articles: (i) research articles; (ii) the second filter eliminates articles that do not consider sustainability or resilience as a whole; (iii) contextualized within the framework of globalization; (iv) English language; and (v) open access. All the articles were classified according to four main components: supply chain, resilience, sustainability, and electric vehicles (electromobility); details are explained later. Figure 2 shows a flow diagram for the scheme for the literature review, which included searches of databases and registers only covering the period from 2001 to 2023. The collected items were organized and administered through Mendeley.

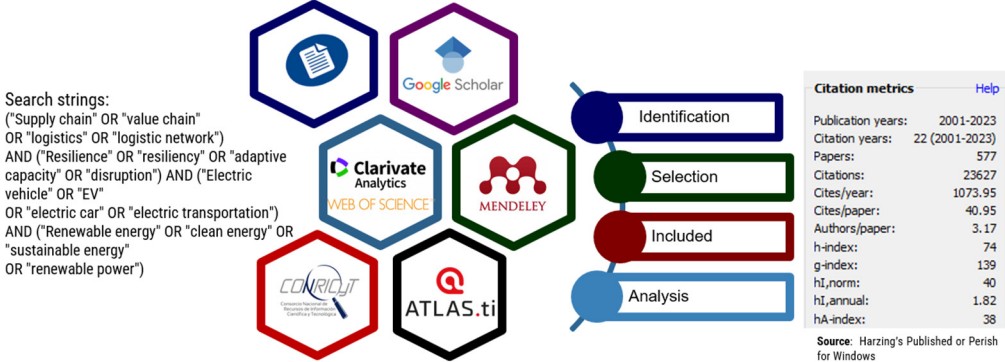

**Figure 2.** Scheme for the literature review. Source: own elaboration from data obtained from the database.

In this review, we followed the guidelines proposed by [17,18], which provide guidance on the literature review and serve as a valuable resource for researchers in the field of organization investigation because the focus is directed toward the research methods and techniques used in the direction and administration of organizations. Additionally, the objective of the review is not to perform a quantitative synthesis of the results through a meta-analysis but simply to identify emerging themes and return to the qualitative evidence.

With the search criteria described in this section, a final sample of 74 articles referring to electromobility, supply chain, and sustainability was identified. Reports were uploaded to Atlas-ti software, 2022 version [18], for coding complete article data and analysis with data extraction forms. Table 1 shows four main categories selected for the information analysis, with a brief description of each and the attributes that can be obtained from the category.

**Table 1.** Categories of analysis for data extraction.

| Category | Description | Attribute |
|---|---|---|
| Organization | Organizational structures are the functions each link performs and are associated with the processes and knowledge necessary to execute each activity. | Strategic level<br>Gestion level<br>Operative Level |
| Processes | Corresponds to the activities carried out in the operation of the different links of the electromobility supply chain with the use of renewable energies. | Efficiency<br>Flexibility<br>Reliability<br>Availability |
| Technology | They are the applications that automate processes, functions, activities, data, operation, location, and classification within the electromobility supply chain. | Connection<br>Connectivity<br>Redundancy<br>Outage Recovery<br>Development of Innovative solutions |
| Information | Indicators that measure the performance of the treasury represent the objectives and perspectives, documents that originate the logical data of the operation, and conceptual elements to carry out risk management. | Cooperation Action Network<br>Diversification Stocks<br>Relocation Actions<br>Traceability and Transparency of Operation |

Source: own elaboration.

When using this rigorous methodology, the literature review contributes to a holistic understanding of the subject, consolidating knowledge about the resilience of the electromobility supply chain and providing valuable information for future research and practical implementation.

Studies on sustainability and resilience management practices in electromobility supply chains with renewable energy found 38 articles; the configuration of collaborative networks in 27 papers; 9 reports identify processes and implementations of a sustainable supply chain; and 9 analyze the factors that intervene in the operation and resilience strategies in the electromobility supply chains [19–21].

## 3. Results

The management based on the functions and responsibilities of the actors involved in the supply chain is a partial vision that does not establish the generation of value of the processes that go from acquiring raw materials to the final product. Therefore, the theoretical approach based on the process-based view of supply chain management focuses on understanding and improving the processes throughout the supply chain. Process-based management recognizes that companies do not operate in a vacuum but are interconnected with suppliers, manufacturers, distributors, and customers. Therefore, it is essential to understand the processes at each chain stage and how they interrelate. By adopting this vision, organizations seek to identify the supply chain's key strategies and critical points to improve efficiency, quality, and responsiveness to risks and operational obstacles [18]. From the sustainability approach, supply chains must meet environmental and social criteria by satisfying customer and economic needs.

Sustainability has been integrated into the center of the discussion of strategic planning, from the theory of resources and capabilities to generate competitive advantage, which has resulted in the management of sustainable supply chains that refers to the control of the flow of inputs, capital, and information, as well as collaboration between organizations throughout the supply chain, while considering the three dimensions of sustainable development (economic, environmental, and social), which arise from the needs of consumers and stakeholders. A key factor is collaboration between organizations, as it indicates the mechanism that makes sustainability management in the supply chain.

Sustainability management in the supply chain is defined as the management of inputs, capital, information flow, and risk control through the collaboration of and between organizations along and across the supply chain while integrating the three dimensions of sustainable development, which are economic, social, and environmental, which are derived from regulations and stakeholder demands [19,22–25].

Sustainability and supply chains present some areas for improvement of organizations in the knowledge about the social and environmental impacts resulting from their operations, as well as the difficult access to data on the entire supply chain and the present limitations of sustainability measurement tools. In other words, incorporating good sustainability practices for supply chain operations [26]. However, managing the supply chain with the integration of renewable energies poses challenges, such as supplying essential materials like lithium, cobalt, and nickel, which come from limited geographical locations. Therefore, recycling initiatives and advanced technologies are being explored to reduce reliance on critical materials and diversify the supply chain, improving sustainability. The supply chain is a complex system constantly changing and evolving in response to environmental changes and internal and external disturbances. Table 1 shows strategies for applying sustainability in supply chain management for specific regions, industries, or countries. However, managing the supply chain with the integration of renewable energies poses challenges, such as supplying essential materials like lithium, cobalt, and nickel, which come from limited geographical locations. Therefore, recycling initiatives and advanced technologies are being explored to reduce reliance on critical materials and diversify the supply chain, improving sustainability.

Assessing potential vulnerabilities associated with the availability and reliability of renewable energy sources is essential to meet the growing demand for electric vehicle charging and factories running on renewable energy. Government regulations and policies play a critical role in ensuring supply chain resilience by promoting the development and stability of renewable energy infrastructure. Since energy is one of the main components for

the development of infrastructure advances, efforts must be oriented not only to the devices that consume energy but also to the generation, distribution, and storage sources [27]; among the main alternatives is the generation of solar energy supported by policies that encourage its adoption. In general terms, distributed generation is a solution that offers low investment, efficiency, and flexibility, thus integrating itself as part of the integral solution for sustainability and electromobility that works through renewable energy sources [28]. The conjunction of renewable energy and electromobility is a win-win for both sectors. It can also lead countries to achieve their goals in the reduction of pollutant gas generation, reliability, and low costs due to two main reasons: (1) the charging of electric vehicles serves as a constant demand that encourages the modernization of the electrical infrastructure; and (2) the flexibility of charging favors low-cost energy such as solar or wind maximizing the use of this energy when the supply is high [29]. There are cases where pricing strategies allow the flexible use of electric vehicles; the greater the use of electric vehicles, the greater the cost reduction [30]. The latent example is the models developed for charging vehicles for public transport; for example, charging in off-duty times (overnight), chargers in terminals, and direct charging on route with the systems (flash charger) [31].

An essential part of the transition to electromobility with renewable energies is the public policies that promote its adoption, vehicle charging, and electric generation infrastructure; promotion of electric vehicles, taxation, taxes for electricity, and renewable energy resources are some examples of the policies considered. Fiscal policies have been found to impede the adoption of electric vehicles, while there are financial incentives that seek their rapid incorporation [32].

Considering that the supply chain is a complex system constantly changing and evolving in response to environmental changes and internal and external disturbances, this theoretical approach is based on the idea that complex systems are systems composed of many interconnected elements that interact and constantly evolve and adapt. In the case of the renewable energy electromobility supply chain, the interlinked elements may include suppliers, manufacturers, distributors, retailers, customers, and other key players.

From the theory of complex adaptive systems, the decision-maker that researches how to majorly improve the supply chain's resilience suggests that it is necessary to encourage adaptation and flexibility in the different supply chain links (Table 2). This can be achieved by creating collaborative networks and promoting innovation and diversification in the supply chain [33]. In addition, it is essential to promote transparency and communication in the supply chain so that the different actors can share information and collaborate in solving problems and making decisions. This third section summarizes the information obtained from the literature review. The main limitation of this paper should be noted. Although the process of searching the literature was followed, it is possible that some papers were not identified.

The supply chain plays an essential role by facilitating the movement of raw materials, capital, and information between the parties involved in producing and consuming goods [34]. Traditional supply chain philosophies have evolved to include social, environmental, and energy-related objectives for competitive advantage. Discussions within modern industrial companies have led to a broader recognition of environmental issues within the general context of organizational challenges and crises [35,36], driving the concept of a sustainable supply chain since the late 1980s.

Sustainability can be seen as a business approach that strives to generate long-term, consistent economic returns while minimizing operations' environmental and social impacts. It emphasizes integrating environmental integrity, economic prosperity, and social equity, with each principle necessary but insufficient [7,19,37]. Organizations have increasingly embraced sustainability in their operational performance, with decarbonization being a key sustainability strategy in electromobility supply chains [4,35,38–40].

**Table 2.** Electromobility supply chain management.

| Ambit | Resilience | It's Not Important | Less Important | Neutral | Important | Very Important |
|---|---|---|---|---|---|---|
| Changes | Orientation to the environment | | | ★(purple) | ★(cyan) ★(green) | ★(yellow) ★(pink) |
| | Learning capacity | | | ★(cyan) ★(purple) | ★(yellow) ★(green) | ★(pink) |
| | Adaptation to change | | | ★(green) | ★(cyan) ★(purple) ★(yellow) | ★(pink) |
| | Emergency planning and recovery | | | ★(purple) ★(green) | ★(cyan) ★(pink) | ★(yellow) |
| | Sustainability Practices | | | ★(green) | ★(purple) ★(yellow) | ★(cyan) ★(pink) |
| Management | Key actors and decision making | | ★(purple) | ★(cyan) ★(green) | ★(yellow) ★(pink) | |
| | Continuity of operations during the crisis | | | | ★(cyan) ★(green) | ★(purple) ★(yellow) ★(pink) |
| | Analysis of information and monitoring requirements | | | | ★(yellow) ★(green) | ★(cyan) ★(purple) ★(pink) |
| | Architecture of the collaborative network and prevention of operational interruptions | | | ★(cyan) ★(green) | ★(yellow) ★(purple) ★(pink) | |
| | Technological innovation | | | | ★(cyan) ★(green) ★(pink) | ★(purple) ★(yellow) |
| Processes | Flexibility | | ★(cyan) | ★(green) ★(pink) | ★(purple) ★(yellow) | |
| | Processes and sustainable production | | | | ★(purple) ★(green) | ★(cyan) ★(yellow) ★(pink) |
| | Fossil fuel consumption conversion | | | | ★(cyan) ★(yellow) ★(green) | ★(purple) ★(pink) |
| | Reducing input waste in production and supply chains | | | ★(green) ★(pink) | ★(purple) ★(yellow) | ★(cyan) |
| | Evaluation indicators | | | | ★(cyan) ★(purple) ★(pink) ★(green) | ★(yellow) |
| CODE | Sustainability ★(cyan) | Fossil energies ★(purple) | Renewable energy ★(yellow) | Supply chain ★(green) | Electromobility ★(pink) | |

Note: take for research's adoption of sustainable practices in the supply chain of electromobility.

The resilience of the electromobility supply chain is an important part of managing a business that covers the acquisition and distribution processes that affect a company's operations. Resistance to the risk of supply chain interruption requires a collaborative and proactive approach to all interested parties. The findings derived from the literature review established the key theoretical concepts shown in Table 2, where the conceptual dimensions and the relationship that unites them are summarized.

The supply chain is the description, planning, and management of materials, information flows, and logistics activities of and between organizations. On the other hand, the supply chain is conceived as the interactions between a group of companies [41,42] within and among themselves [43] to satisfy a consumer [44] through a good or service [45] whose dynamics and interconnection tends to the automation [46] of information and communication technologies and diverse software [47], moving from horizontal composition [48] to network configuration [49], collaboration [50] to take advantage of the opportunities provided by production environments and platforms with interfaces between companies in the design of customized solutions for each client. Both approaches to the concept of the supply chain assume that the links are managed by their different actors using technology and collaborative networks in an environment of information, product, and service exchange [51].

## 4. Discussion

Since the 1980s, the term supply chain has been associated with sustainability with the combination of functions, processes, and relationships supported by information and financial transactions that are mobilized within and between companies from manufacturing, transporters, warehouse workers, retailers, and final consumers referred to as supply chain [52,53].

### 4.1. Integration of Sustainability Practices into the Supply Chain

Authors such as [54] classify the demand of the various stakeholders into three types: best sustainability practices, sustainability indicators, and the relationships between the multiple stakeholders. The significant impacts of the rules are waste elimination, supply chain risk management, and cleaner production [55]. On the other hand, it states that the leadership and sustainable philosophy manifested in the governing bodies within an organization [56,57] are aspects that promote sustainable best practices.

Thus, global awareness of environmental and social problems has led governments, business organizations, and various actors to promote the adoption of sustainable practices with a triple approach [58]. The supply of lithium depends on a high geographic concentration, and an oligopolistic market structure is vulnerable to unexpected changes that guarantee the transition to sustainable electromobility in the long term. This is attributed to lithium's availability, lithium-ion batteries' recycling, and environmental degradation [59].

### 4.2. Collaborative Networks

The growing demand for sustainable products originated from an upward trend of responsible consumption [60,61]. Consequently, the development of sustainable goods and services implies the commitment of all members downwards with suppliers and upwards with customers of the supply chain, suggesting that the configuration of collaborative networks that contribute with inter-organizational dynamics to the strengthening of resources and knowledge transfer capabilities, the shaping of solutions and the motivation of the activity for collaboration is a crucial factor for the effective and efficient development of a sustainable supply chain.

Inputs and outputs [56] refer to the supply chain's resources and inputs derived from managed capital, information, and materials [19]. Resources are identified as non-renewable–renewable and non-recyclable–recyclable [62], and the lack of availability of such resources increases the emphasis on renewable and recyclable resources. Reinserting recyclable, repaired, or recovered resources back into their chain of origin can add value [63].

The flow of information between the organization and its direct supply chain in the process requires practices associated with labor rights and environmental performance from inputs to outputs to manage risks [19,64–66].

### 4.3. Processes

Sustainability in the supply chain is defined as a process [67], which is why the externalization of practices depends on the relationship between organizations with other stakeholders, in addition to the type of demands they receive from the latter [60,68]. Hence the relevance of the creation of environmental and social performance manuals and standards [60] that formulate indicators of compliance with established objectives such as reverse logistics and remanufacturing [63], green specifications, purchasing, and logistics [56,69,70], product stewardship, and operations planning [68,69,71–73]. In addition, the adoption of new alternatives to mobility through electric energy in the search for the reduction of air pollution with a view to future trends and new restrictions on the use of conventional vehicles [1,3,74].

### 4.4. Concept of Resilience in the Electromobility Supply Chain

The concept of resilience in the electromobility supply chain encompasses several key aspects:

■ Flexibility: a resilient supply chain is flexible and adaptable, capable of responding and adjusting to changes in demand, disruptions in the availability of critical materials or components, and shifts in market conditions;
■ Redundancy: resilience often involves having redundant or backup systems, resources, and processes in place to prevent single points of failure;
■ Risk assessment and management: resilience requires a proactive approach to identifying and assessing potential risks and vulnerabilities within the supply chain;
■ Collaboration and coordination: resilience is enhanced through stakeholder cooperation and coordination throughout the supply chain;
■ Continuity planning: developing robust business continuity plans is essential for resilience;
■ Integration of advanced technologies and data-driven approaches: real-time data monitoring, predictive analytics, and automation to enhance visibility, traceability, and agility within the supply chain.

The electromobility supply chain's resilience aims to minimize disruptions' impact, maintain a steady flow of materials, components, and products, and ensure the sustainable growth and performance of the electromobility sector [75,76]. By building resilience, the supply chain can effectively navigate challenges, recover quickly, and continue supporting electromobility adoption and expansion [77], also known by the term intelligent supply chain, integrating 4.0 technologies in supply chain management [78]. An example of big data monitoring is the coordination of a more extensive fleet of self-driving vehicles developed by the arrival of COVID-19 using algorithms for the system to make decisions in real time.

There are a number of key characteristics for resilience that have been applied even for cities: efficiency—doing more with less is part of any system to carry out an adequate use of resources; connectivity—having proper communication between different actors is vital to carry out recovery actions together; inclusiveness—facilitating participation between communities and institutions for a free movement of resources for the recovery period; diversity—corresponds to the availability of various options to deal with unexpected eventualities; foresight—any system considered resilient must be able to anticipate risk situations that may arise in the future; equity—the principle of equity corresponds to the fair distribution of available resources; creativity—the emergence of solutions or responses should not be conditioned by technology or resources, imagination can be a great tool; adaptability—constant changes can cause disturbances to systems, rapid adaptation to change will bring improvements to the design for the following events; redundancy—a

system that contains component backup can compensate for failures caused by eventualities to disrupt its operations; and robustness—a system that is created and managed appropriately for the conditions it will face will have a better chance of withstanding the stress of unexpected situations [15,79]. The COVID-19 pandemic was a global event demonstrating how vulnerable supply chains can be, testing their resilience regarding demand, supply, changes in customer behavior, and transportation requirements. Learning from a situation like this demonstrates that chain capabilities can be improved; for example, by decentralizing chain activities, diverse suppliers, cooperative action networks, and being out of touch with market needs [80,81]. An example of actions taken in the resilience of supply chain management is covered inventory. Having a safety stock is a strategy to increase resilience. This involves holding an additional inventory level to cover potential delivery delays or product shortages if this does not consider an extra cost for the company. If there is a supply disruption, the company can use the safety inventory to maintain its operations until normalcy is restored. During COVID-19 many industries suffered from disturbances in their supplies, not only of raw materials but also of processed products essential to produce complex items. The automotive industry, for example, continues to suffer from a shortage of semiconductor materials to build vehicle control computers.

Figure 3 briefly outlines the points necessary for the supply chain to recover from unexpected events that result in performance impacts. It complies with six key points to achieve resilience. Regarding some theories on sustainable supply chain, two can be mentioned: a practice-based view (PBV) is based on the observation of the set of activities that take place in the supply chain, where the joint action will serve as a connection with the environmental strategies established by the decision-makers to create networks and improve the use of resources that will be reflected in the improvement of performance.

Practice-based studies (PBS) is based on explaining what the practice refers to and how it does it, to establish a dynamic routine. The contribution of this theory is centered on the fact that the study of nature and sustainability facilitates its explanation in the supply chain, in addition to highlighting that it is necessary to understand the practices of the chain. In this context, organizational learning is mainly affected [82].

The links of the supply chain electromobility must be managed by its different actors using technology and cooperation networks, within the framework of the 4.0 revolution, in an environment of exchange of information, products, and services. In short, the systems interact with each other in real time with the support of big data, cloud computing, artificial intelligence, industrial automation, new processes, and decision making with a focus on product innovation and techniques so that value is generated for society. One of the most significant examples is the dependence on minerals in the electromobility supply chain due to the need for efficient batteries that require graphite, lithium, and cobalt [83].

In the era of digitalization, the resilience of the electromobility supply chain is supported by solid business continuity plans that integrate technological instruments to guarantee that each link contributes to the balance between social, ecological, and economic well-being but, at the same time, can satisfy present needs without compromising future needs. Technology integration has considered incorporating sustainability in the design and implementation of charging infrastructure for electric vehicles, as it is known the environment is supported by engineering with innovative solutions. Hence, the energy sector is critical to electromobility; the circle produces energy-efficient devices, generates renewable energy, and closes with economic energy use [84].

Figure 4 shows seven critical points for electromobility supply chain management processes. After synthesizing the literature that addresses the sustainable management of the supply chain and identifying electromobility trends. The theoretical foundations of the conceptual model of the resilience of the sustainable supply chain.

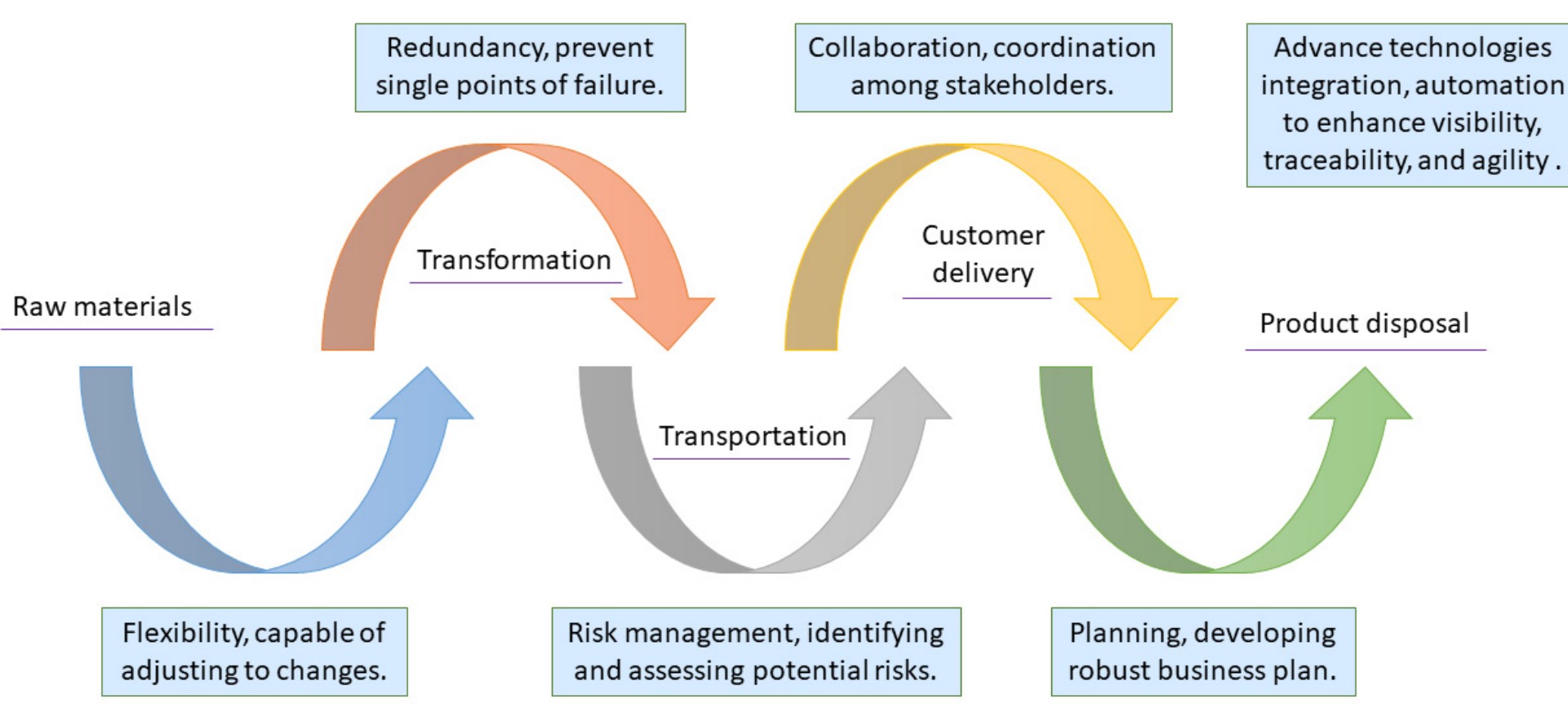

**Figure 3.** Resilience in supply chain.

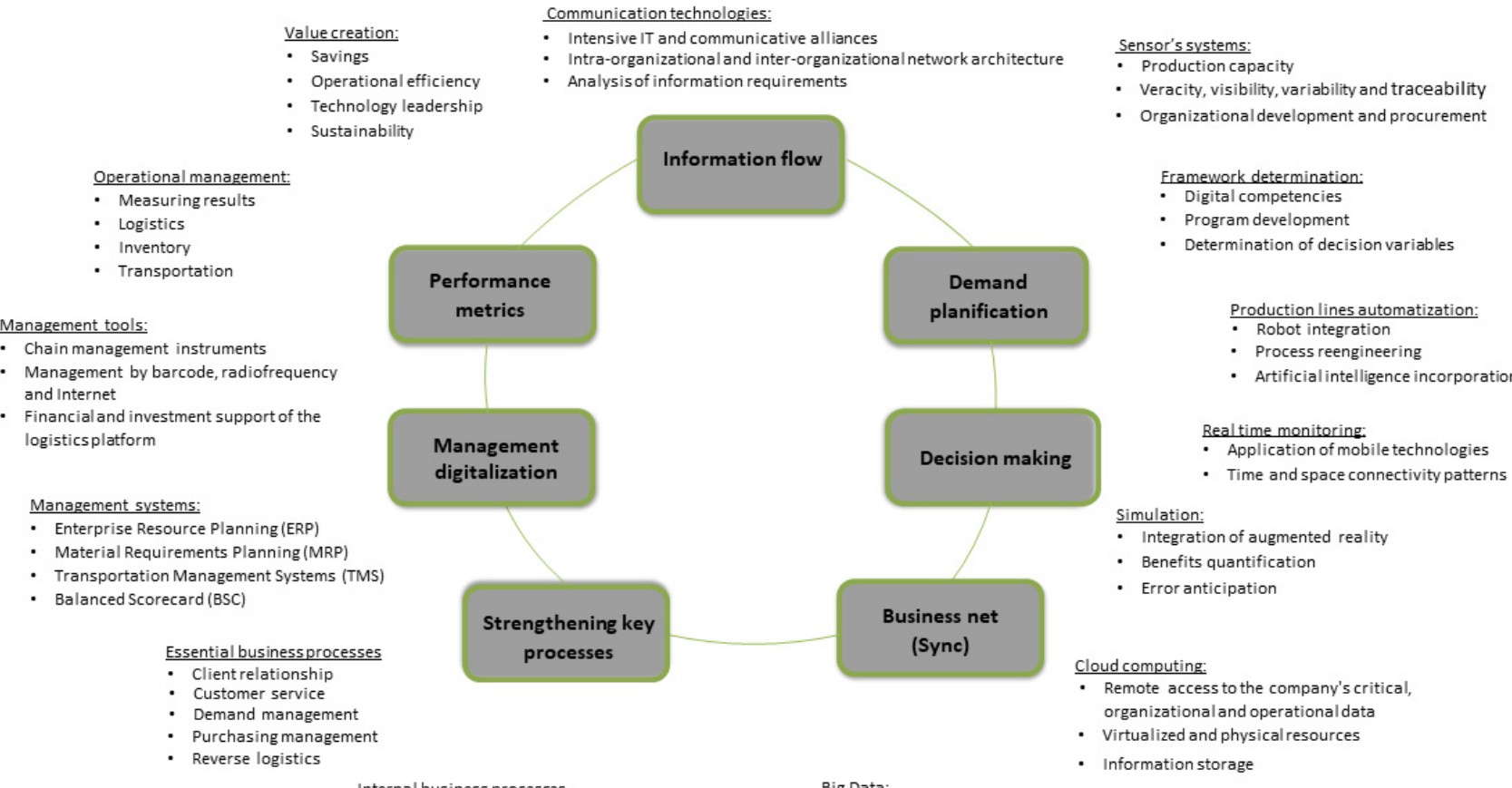

**Figure 4.** Resilience supply chain management.

The information flow of electromobility supply chain management aims to coordinate the parties' operations by transferring data and knowledge that can become useful for the process of the entire chain. Technology, communication, and networks, not only within but also outside the organization, play an essential role in this process because the information is generated from acquiring raw materials until the customer receives the finished product [85]. Demand planning seeks to estimate and forecast the organization's needs in a certain period [86]. The production capacity, the organization's development, and the establishment of decision variables will make it possible to establish a frame of reference that will serve as a tool for making informed decisions on inventory, distribution, production, and raw materials to satisfy customers' needs efficiently.

Decision making in electromobility supply chain management is a critical point that aims to select the best alternative to achieve the best performance of the whole chain. The primary foundation for this point is the simulation and analysis of data present from the strategic plan to the operations at different levels. This point requires technology and tools that allow modeling and predicting the chain's behavior based on previous data that serve as a basis for the decision-maker [76].

The networking and synchronization of the business go hand in hand with strengthening key processes. Since we are talking about supply chain management, all actors are under the same conditions, so their objective is to coordinate and collaborate actions that optimize the flow of materials, resources, and information [87]. The strength of the supply chain will lie in the capabilities of the link with the most significant difficulties.

The digitization of electromobility supply chain management seeks to increase the efficiency and transparency of the chain using technologies and tools applied to management systems such as ERP, MRP, TMS, and BSC. To carry out the proper digitization of the supply chain, it is necessary to be able to exchange data electronically, which involves process automation, data analysis, and incorporating the Internet of Things (IoT) and artificial intelligence (AI) to not only increase process optimization but also to detect errors or fraud to be incorporated into risk management [88].

Lastly, there are performance metrics that will make it possible to evaluate the performance of activities related to the chain. The advantage of having performance metrics is that the manager can visualize the information in a synthetic way and with measurable data, eventually guiding the entire chain to improvement [89]. A wide variety of variables can be considered for improvement metrics, such as customer service level, inventory turnover, raw material consumption, transportation efficiency, and logistics cost. In this way, the supply chain management process is fulfilled so that it can be fulfilled again and be part of the continuous improvement process.

Supply chains per se make up the operation of the global economic processes and originate many business opportunities, as well as lead to the identification of social and environmental impacts, intrinsically dynamic and complex, facing the challenge of the scarcity of non-renewable resources faced by organizations in the present and in the future, which requires the need for the adoption of more sustainable practices from a three-pronged approach. Therefore, the trend is to increase the depth of sustainability management in the supply chain as an integral object of study.

The supply chain's resilience is the ability to adapt to changing circumstances, maintaining essential functions and operations while facing interruptions with irrigation, complex scenarios, high levels of complexity, and uncertainties. In the context of the electromobility supply chain, its resilience will be determined by the agility and flexibility of the production systems and the suffering and associated infrastructure batteries, as well as the implementation of sustainable practices and the adoption of advanced technologies for real-time data monitoring, it is the traceability of operations with a predictive approach that improves the visibility of possible interference, interruptions, threats, and risks. For example, in the case of electromobility using clean energy that has a high dependence on the storage sources of accessibility to materials such as nickel, graphic, lithium, cobalt, semi-

conductors, and power electrons which are essential for the batteries of electric vehicles for autonomy and operation. This implies having support systems to prevent supply failure.

Another challenge is the management of toxic waste associated with the short period of the life of batteries. Therefore, proposals are required to extend your helpful life with a proactive approach to evaluate risks and vulnerabilities within the supply chain. In this sense, the collaboration and coordination of interested parties for developing continuity plans for operating business models in the circular economy are necessary.

An extra important part of resilience in the sustainable electromobility supply chain is the technological advancement of electric propulsion machines. Machine construction has long been carried out using permanent magnets, a reality that is changing to seek efficiency and revolutionary designs that meet the high ethical standards set by the industry [90]. Several measures can be taken to improve the resilience of the electromobility supply chain. These include diversifying the sources of materials to reduce supply and supply dependence concentrated in specific regions and exploring alternative materials, and recycling initiatives to minimize the shortage of materials and develop sustainable elimination practices for batteries. In addition, integrating renewable energy sources in the supply chain depends less on fossil fuels and reduces greenhouse gas emissions. However, guaranteeing the availability and reliability of renewable energy sources also becomes a consideration to evaluate the supply chain's resilience.

Charging networks and infrastructure, together with electromobility, are the key to the decarbonization of transportation; the availability of the connection between generation networks and vehicle battery banks will increase the adoption of these new technologies [91]. Lithium batteries, for example, face a series of problems such as price instability due to political differences, constant increase in the demand for the mineral, and the life cycle of the batteries, so actions such as the creation of models to predict the rise in the number of electric vehicles, the final disposal of these vehicles and the increase in the need for minerals would be beneficial. In addition to strategic alliances between governments, manufacturing companies, and academics that allow the correct use of waste for the environment and an uninterrupted supply for the industry [92]. The automotive sector in the United States has transformed over time, driven mainly by great competition and the need to be at the forefront; between 2022 and 2025, 50 plants will produce and will produce electric vehicles. However, these production plants will continue manufacturing internal combustion cars [93]. Vehicle production projections show that the United States market is transitioning to a transformation driven by the costs of electric vehicles below those of gas-powered cars [94].

## 5. Conclusions

In response to the main questions that governed the literature review and in compliance with the objective of this research, it was found that the most significant impacts of the best practices applied in the supply chain are waste reduction, cleaner production, transition to new energies in mobility, and risk management, and that the leadership and sustainable philosophy manifested in the governing bodies of an organization are aspects that promote the best sustainable practices. Furthermore, the orientation towards sustainability requires the effort and commitment of all stakeholders, which suggests that the configuration of cooperation networks implicit in the organizational dynamics is a critical factor for the effective and efficient development of a sustainable supply chain, as well as the use of renewable energies in the areas of production and transportation, which can add value to the entire chain.

Sustainability, and its critical interfaces with supply chain management electromobility, suggest that the primary way to do business in the 21st century is to have professionals and policymakers design and implement sustainable practices in the configuration of collaborative networks of the various actors involved in supply chain management. Future research needs to be conducted in practical application to be contrasted with reality in a

defined social framework to investigate how collaboration that adopts sustainable practices is articulated.

The trend in adopting Industry 4.0 technologies from a critical vision of technological innovation with the introduction of different applications and tools in the functions, processes, and tasks reconfigure the supply chain and favor the intervention of other interest groups in continuous improvement processes. The findings of the study highlighted that, in line with the achievement of sustainable electromobility with the use of renewable energies, it is necessary to implement strategies of collaboration and participation of the actors for the effective use of the components (systems, modules, or complete battery cells), by extending their useful life through reuse or remanufacturing and subsequent efficient recycling.

In summary, this document has outlined different challenges and included the description of some strategies for its sustainable management and increasing its response capacity over time to possible events and risks derived from the operation of the links that comprise it, in addition to providing a model of the electromobility supply chain with renewable energies, placing the use of digital technologies and platforms at the center of its management for its continuity. From here, it is proposed from the strategic field that the participation of the actors goes beyond the interaction with the different interest groups and covers a set of actions, such as collaboration in the design of continuity plans, the development of local supply and integration of digital innovations typical of smart mobility executed from the ethical principles of social well-being, and environmental justice in a globalized society.

Therefore, studies on resilience in the electromobility supply chain suggested that the impact of climate change and risk management cannot be separated in the global social and economic context of the territorial space. In this way, incorporating the territorial dimension can help close the gap between threat and vulnerability and find ways to make sense of adaptation as a profound transformation.

**Author Contributions:** Conceptualization, A.D.T.-R. and A.d.J.M.N.V.; methodology, A.D.T.-R.; formal analysis, R.F.D.S., A.D.T.-R. and A.d.J.M.N.V.; investigation, A.D.T.-R., A.d.J.M.N.V. and R.F.D.S.; writing—original draft preparation, A.D.T.-R., A.d.J.M.N.V. and R.F.D.S.; writing—review and editing, A.D.T.-R. and R.F.D.S.; supervision A.D.T.-R.; project administration, A.D.T.-R. and A.d.J.M.N.V. All authors have read and agreed to the published version of the manuscript.

**Funding:** This research received no external funding, but the APC was partially covered by Instituto Politécnico Nacional.

**Institutional Review Board Statement:** Not applicable.

**Informed Consent Statement:** Not applicable.

**Data Availability Statement:** Not applicable.

**Conflicts of Interest:** The authors declare no conflict of interest.

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
