# Peer review of "The Resilience of the Renewable Energy Electromobility Supply Chain: Review and Trends"

_sustainability, doi:10.3390/su151410838_

Round 1

Reviewer 1 Report

The authors presented a (systematic?) literature review of the resilience of the Electromobility supply chain. While the review is good, I identified the following shortcomings in the paper. 

1.     You state in the title that you conducted a systematic literature review. However, you don’t report the results as it would be expected for a systematic literature review, nor is the method you describe how a systematic literature review is conducted. If you have not conducted a systematic literature review, please remove this from the title and the text. If you have conducted a systematic literature review, please describe the methodology and present the results following the PRISMA guidelines. 

Minor:

2.     What happened to the references, you seem to mix styles:  (Abbasi & Nilsson, 2012; Ahi & Searcy, 2013; 268 Min & Kim, 2012; [15, 7, 27].

3.     I personally would recommend splitting the discussion and the results instead of combining them into one section. As both have a different focus. If you can, try to structure these sections a bit more clearly. 

4.     You could have a look at similar literature reviews and could compare the findings of your literature review with the findings of these literature reviews.

5.     If you can, it might be good to highlight a bit better which part of the paper describes table 1. You cite it a few times in the text before the table. But it seems as if you describe it as well after the table without citing it. 

Author Response

Greetings,
I thank you for the comments derived from your review. Attached is a detailed list of the actions carried out to meet your recommendations.
Sincerely
Alma Delia

Reviewer 2 Report

First of all, I appreciate the opportunity to review the paper Electromobility supply chain resilience with renewable energy: a systematic review of literature.  The paper deals with very interesting problem.

Suggestions are below:

·        Title formulation: change „a systematic review of literature“ with systematic literature review. SLR is special type of review paper and has unique methodology.

·        The abstract is not well written. The most important results must be listed.

·        Keywords should include: review, literature review, or SLR.

·        Methodology section is poorly written and not acceptable. There is no SLR methodology. This kind of paper must have a very clear methodology (journals, keywords, databases). There are numerous papers for methodology (see Suggested References).

·        Why such a short period 2018-2023?! Are there review papers in the same area which covers period before 2018.

·        The review paper should not just be a list of what everyone has done but should identify trends and gaps in the literature and offer suggestions for furthering the field relative to the specific phenomenon, with a VERY STRONG CRITICAL VIEW AND VERY STRONG METHODOLOGY.

·        It is necessary to understand the purpose and aim of the paper as well as its "position" in relation to previous research (also gap analysis).

·        The separate section Practical and theoretical implications (or Discussion) is missing. The existing section Discussion is very modest. This confirms the lack of scientific and practical contributions.

·        The paper is descriptive and analytic, not critical and exploratory.

·        The paper lacks scientific research rigor, the research steps are not systematic and objective.

·        The separate section Practical and theoretical implications (or Discussion) is missing. This is very important for the scientific and practical contribution

·        Conclusion section is not on a satisfactory level. The conclusion in scientific papers is very important.

o   Limitations of your research must be emphasized

o   Future research directions are missing.

o    

Suggested References

Denyer, D. & Tranfield, D., (2009). Producing a systematic review. In D. Buchanan & A. Bryman (eds.) The sage handbook of organizational research methods. Sage Publications Inc., Thousand Oaks, CA, 671-689.

Kilibarda, M., Andrejić, M., & Popović, V. (2020). Research in logistics service quality: a systematic literature review. Transport, 35 (2), 224-235.

.

Author Response

Greetings,
I thank you for the comments derived from your review. Attached is a detailed list of actions to meet your recommendations.
Sincerely
Alma Delia

Reviewer 3 Report

This paper focuses on the resilience of the electromobility supply chain and how it can integrate renewable energy sources, however there are some questions should be discussed carefully.

1. The paper should introduce more current and future trends, such as the development of renewable energy and electric vehicle technologies, to support the discussion on sustainable supply chains.

2. When discussing the resilience of the electric vehicle supply chain, further exploration can be done on the application of advanced technologies and data-driven methods such as real-time data monitoring, predictive analytics, and automation to enhance the visibility, traceability, and agility of the supply chain.

3. To better explain the concept of resilience in sustainable supply chains, more explanation and specific examples can be provided.

4. The concept and development of sustainable supply chains are mentioned in the paper, but there is no further explanation of this concept or citation of relevant theories. It is recommended to provide a more comprehensive explanation of the concept of sustainable supply chains in the paper and cite relevant theories to support the discussion.

It is OK.

Author Response

(The authors gave the same response as above.)

Round 2

Reviewer 2 Report

Unfortunately, the authors did not take the opportunity to improve the paper. This is not a review paper but a review of literature for a more serious paper/research.

The entire response is based on the assumption that the paper is not SLR. However, all types of review papers must have all mentioned parts. This is not a scientific paper.

The author's responses are not serious or in the manner of scientific research.

FOR EXAMPLE:

“”””””””””””””””””””””””””

Point 3: Keywords should include: review, literature review, or SLR.
Response 3: The title was changed, so it no longer applies to include a systematic review of the
literature.

THE WORD REVIEW IS IN THE TITLE AND THE PAPER TRY TO BE REVIEW! THIS IS KEY WORD.

Point 4: Methodology section is poorly written and not acceptable.
Response 4: The title was modified, since it is not a systematic review of the literature, therefore, the methodology of the systematic review of the literature is not included.

?

!?!?! ALL PAPERS ARE MUST-HAVE METHODOLOGY!?

Point 5: Why such a short period 2018-2023?
Response 4: It is not a systematic review of the literature, therefore, we only worked with the
publications of the indicated period.

WHY?

Point 6: The review paper should not just be a list of what everyone has done but should identify
trends and gaps in the literature and offer suggestions for furthering the field relative to the specific
phenomenon, which a very strong critical view and very strong methodology.
Response 6: It is not a systematic review of the literature, therefore, we only worked with the
publications of the indicated period

IS THIS REALLY YOUR ANSWER TO THE POINT 6!??

Point 7: It is necessary to understand the purpose and aim of the paper as well as its “position” in
relation to previous research (also gap analysis).
Response 7: The title was adjusted to the purpose and objective of the document, the gaps
were analyzed based on the proposed conceptual model of the electromobility supply chain

THE MAGIC ARGUMENT "WE CHANGE TITLE"!?

Point 8: The separate section practical and theoretical implications (or discussion) is missing. The
existing section discussion is very modest. This confirms the lack of scientific ad practical
contributions.
Response 8: The electromobility supply chain is a developing field that arises within the
framework of smart cities; therefore, its theoretical foundation is under construction, and
there is little empirical evidence.

THE ARGUMENT IS INVALID!

Point 10:
objective.

The paper lacks scientific research rigor; the research steps are not systematic and

Response 10: The development of the study does not respond to a systematic review of the
literature, but to develop the research, specialized databases were consulted that served to
trace the development of the analysis.

Point 10: The separate section Practical and theoretical implications (or Discussion) is missing. This is very important for the scientific and practical contribution

Response 11: equal the point 7

””””””

.

Author Response

Dear Reviewer 2, we appreciate all the suggestions. We hope the point-by-point response to the comments is satisfactory because it considers these corrections.
Greetings

Reviewer 3 Report

The reviewer's comments have been meticulously responded and the revised manuscript is satifactory. I have no furture comments.

Author Response

Dear Reviewer

We want to express our sincere thanks for your recommendations and suggestions. We sincerely appreciate your time and effort in carefully reading and evaluating the article. Your constructive feedback has been of great help in identifying opportunities for improvement.

Once again, we want to express our appreciation for your time, dedication, and shared expertise.

We warmly greet you,

Alma, Angel, and Rodrigo.

Round 3

Reviewer 2 Report

The paper should be accepted for publication.

.